# Estimating conditional density of missing values using deep Gaussian mixture model

**Marcin Przewięźlikowski** [1]  **Marek Śmieja** [1]  **Łukasz Struski** [1]

## Abstract

We consider the problem of estimating the conditional probability distribution of missing values given the observed ones. We propose an approach, which combines the flexibility of deep neural networks with the simplicity of Gaussian mixture models (GMMs). Given an incomplete data point, our neural network returns the parameters of Gaussian distribution (in the form of Factor Analyzers model) representing the corresponding conditional density. We experimentally verify that our model provides better log-likelihood than conditional GMM trained in a typical way. Moreover, imputation obtained by replacing missing values using the mean vector of our model looks visually plausible.

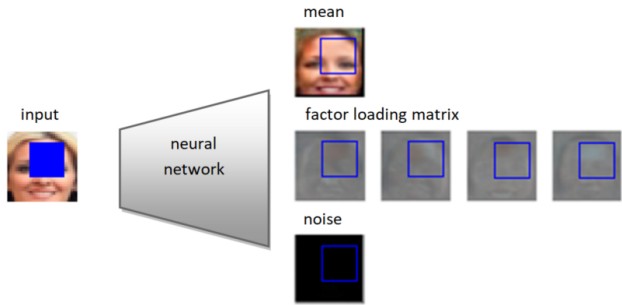

*Figure 1.* The idea of the proposed DMFA. Given a missing data point, our model returns the parameters of conditional Gaussian density: mean, factor loading matrix (we use 4 latent factors) and noise matrix for the model of Factor Analyzers, which describes a distribution of missing data (the area inside the blue square).

## 1. Introduction

Estimating missing values from incomplete observations is one of the basic problems in machine learning and data analysis (Goodfellow et al., 2016). A typical approach relies on replacing missing values with a single vector based on available information contained in observed inputs (Jerez et al., 2010; Van Buuren, 2018). While imputation techniques are frequently used by practitioners, they only give point estimate instead of a probability distribution. Quantifying the probability distribution of missing values plays an important role in generative models (Li et al., 2019a), uncertainty prediction (Ghahramani & Jordan, 1994), recommender systems (Ma et al., 2018) as well as is useful in applying classification models to incomplete data (Śmieja et al., 2018; Williams & Carin, 2005; Dick et al., 2008).

While deep generative models such as VAE, GAN or WAE (Kingma & Welling, 2014; Goodfellow et al., 2014; Tolstikhin et al., 2017) are capable of modeling the distribution

of complex high dimensional data, such as images, it may be difficult to use them to estimate the uncertainty contained in missing data (Mirza & Osindero, 2014; Sohn et al., 2015). In the case of VAE, the nonlinear form of decoder makes the conditional distribution of missing data hard to assess (Nazabal et al., 2020). By applying a type of Gibbs sampling (Rezende et al., 2014; Mattei & Frellsen, 2018) or importance sampling (Mattei & Frellsen, 2019), it is at least possible to generate imputations from this conditional distribution. Similarly, it is challenging to obtain a closed-form expression for such a conditional distribution in GAN, but one can design a procedure to sample from it (Yoon et al., 2018; Li et al., 2019a). It was recently shown that deep flow models can be trained to represent a conditional density as a neural network transformation of some prior distribution (Trippe & Turner, 2018; Li et al., 2019b). Nevertheless, the constructed density cannot be maximized analytically. One can only produce samples or attempt to maximize the corresponding density numerically.

In the case of shallow density models, such Gaussian mixture models (GMMs), we can easily calculate a conditional density function related to missing values in a closed-form (Ghahramani & Jordan, 1994; Delalleau et al., 2012) as well as to maximize it analytically. Moreover, simple Gaussian form of the conditional density function allows us to

---
[1]Faculty of Mathematics and Computer Science, Jagiellonian University, Łojasiewicza 6, 30-428 Kraków, Poland. Correspondence to: Marek Śmieja <marek.smieja@uj.ed.pl>.

*Presented at the first Workshop on the Art of Learning with Missing Values (Artemiss) hosted by the 37th International Conference on Machine Learning (ICML). Copyright 2020 by the author(s).*

combine conditional GMM with other machine learning techniques that can process missing data without using any imputations at preprocessing stage (Śmieja et al., 2018; 2019). Another related line of work has explored autoregressive models for conditional data generation or density estimation (Van den Oord et al., 2016; Papamakarios & Murray, 2016).

In this paper, we propose DMFA (Deep Mixture of Factor Analyzers) for estimating the probability density function of missing values, which combines the features of deep learning models and GMMs. We construct a neural network, which takes an incomplete data point and returns the parameters of Gaussian density (represented as Factor Analyzers model) modeling the distribution of missing values, see Figure 1. Since the proposed network returns an individual Gaussian density for every missing data point, its expressiveness is higher than using GMM with a fixed number of components. In contrast to classical GMM, which is a type of generative model that estimates a density of the whole data, DMFA follows a discriminative approach and directly maximizes the likelihood function on missing values (Ng & Jordan, 2002). In consequence, the obtained Gaussian density has a better quality in the context of missing data than the conditional distribution computed from GMM (see experimental section). Nonetheless, our model still provides an analytical formula for a distribution of missing values, which is useful in diverse applications, and may be more attractive than adapting deep generative models to the case of missing data. We also verified that the imputations obtained by replacing missing values with the mean vector of returned Gaussian density look visually plausible.

Our work is strictly related with (Bishop, 1994), but instead of using isotropic covariance matrix and many Gaussian components for conditional density, we follow (Richardson & Weiss, 2018) and employ Factor Analyzers model, which suits better to high dimensional spaces such as images. Our preliminary work suggests that isotropic covariance is not able to model dependencies between pixels while the mixture often tends to collapse to a single Gaussian.

## 2. Density model for missing data

In this section, we introduce DMFA model. First, we recall basic facts concerning GMM and MFA in high dimensional data. Next, we show how to compute conditional density from GMM. Finally, we present the proposed DMFA – a deep learning model for estimating conditional Gaussian density on missing values.

**Conditional Gaussian mixture model for high dimensional data.** GMM is one of the most popular probabilistic models for describing a density of data (McLachlan &

Peel, 2004). A density function of GMM is given by

$$p(x) = \sum_{i=1}^{k} p_i N(\mu_i, \Sigma_i)(x),$$

where $p_i > 0$ is the weight of $i$-th Gaussian component with mean vector $\mu_i$ and covariance matrix $\Sigma_i$ (we have $\sum_{i=1}^{k} p_i = 1$). Given a dataset $X \subset \mathbb{R}^n$, GMM is estimated by minimizing the negative log-likelihood:

$$l(x) = -\sum_{x \in X} \log p(x).$$

While theoretically GMM can be estimated using EM or SGD, this procedure may fail in the case of high dimensional data, such as images. Observe that for color images of size $32 \times 32$, the covariance matrix of a single component has $4.7 \cdot 10^6$ free parameters. In training phase, we need to store and invert these covariance matrices, which is computationally inefficient and may cause many numerical problems (Richardson & Weiss, 2018).

It is widely believed that high-dimensional data, such as images, are embedded in a lower-dimensional manifold and using full covariance matrix may not be necessary. For this reason, it is recommended to use the Mixture of Factor Analyzers (MFA) (Ghahramani et al., 1996) or Probabilistic PCA (PPCA) (Tipping & Bishop, 1999), in which every Gaussian density is spanned on a lower-dimensional subspace. In contrast to the typical GMM, the covariance matrix in MFA is given by

$$\Sigma = AA^T + D,$$

where $A = A_{n \times l}$ is a factor loading matrix, which is composed of $l$ vectors $a^1, \ldots, a^l \in \mathbb{R}^n$ such that $l \ll n$, and $D = D_{n \times n} = \mathrm{diag}(d)$ is a diagonal matrix representing noise[1] defined by $d \in \mathbb{R}^n$. The set of vectors $a^i$ defines a linear subspace, which spans a Gaussian density $N(\mu, \Sigma)$, while adding a noise matrix guarantees that $\Sigma$ is invertible. The use of MFA drastically reduces the number of parameters in a covariance matrix as well as avoids problems with inverting large matrices. Recent studies show that MFA can be effectively estimated from image data and is able to describe a higher spectrum of data density than GAN models, see (Richardson & Weiss, 2018) for details.

It is important to note that GMM can not only describe a density of data, but is also useful for quantifying the uncertainty of missing data. A missing data point is denoted by $x = (x_o, x_m)$, where $x_o$ represents known values, while $x_m$ describes absent attributes. Given a missing data point $x$, a natural question is: *what is the distribution of missing values given the observed ones?* In the case of density models, the

---

[1]PPCA uses spherical matrix $D$.

answer is given by a conditional density (Ghahramani & Jordan, 1994):

$$p(x_m|x_o) = \frac{p(x_o, x_m)}{p(x_o)} = \frac{p(x)}{p(x_o)}.$$

In contrast to many deep generative models, e.g. GANs or VAE, the formula for conditional density can be found analytically for GMM (see (Colquhoun & Hawkes, 1995) for detailed formula). Note however that in high dimensional spaces, the conditional Gaussian mixture model reduces numerically to a single Gaussian density. This is caused by the fact that the tails of Gaussian densities decrease exponentially and, in consequence, a single component dominates the others (other components simply become irrelevant).

**Deep conditional Gaussian density for missing data.** An important advantage of GMM is that the conditional densities can be calculated and maximized analytically, which may be appealing in the context of missing data. However, GMM is not trained to estimate a density of missing data – its objective is the log-likelihood computed on all data points. In consequence, there are no guarantees that the resulting conditional density gives optimal log-likelihood for missing values.

In this paper, we are motivated by typical deep learning models used for image inpainting (Pathak et al., 2016; Iizuka et al., 2017; Yu et al., 2018), which are trained to give the most reliable imputations. The proposed DMFA creates a Gaussian density, which minimizes the negative log-likelihood on missing values. More precisely, given a data point $x \in \mathbb{R}^n$, we first generate a random binary mask $M$ to simulate missing attributes. The pair $(x, M)$ induces a missing data point $(x_o, x_m)$. DMFA defines a neural network $f$, which takes $(x_o, x_m)$ together with $M$ and returns the parameters of conditional Gaussian density $p(x_m|x_o)$. Following MFA model, we represent covariance matrix using factor loading matrix $A = A_{n \times l} = (a^1, \ldots, a^l)$, and the noise matrix $D = D_{n \times n} = \mathrm{diag}(d)$. In the case of images, $f$ simply returns the mean image $\mu$ and the covariance matrix $\Sigma = AA^T + D$ represented by $l$ images spanning a Gaussian density supplied with the noise image ($l + 2$ images in total) .

Given the output $\mu$ and $\Sigma$ of the neural network $f$, we define a conditional Gaussian density as

$$p(x_m|x_o) = N(\mu_m, \Sigma_{mm}),$$

where $\mu_m$ and $\Sigma_{mm}$ denote the restrictions of $\mu$ and $\Sigma$ to missing coordinates, see Figure 1 for illustration. Since the number of missing values can be different for subsequent data points, $f$ has to output the parameters of $n$-dimensional Gaussian density $N(\mu, \Sigma)$. However, $N(\mu, \Sigma)$ does not need to estimate a density of the whole data. In our case,

$\Sigma_{mm} = A_{m \cdot} A_{m \cdot}^T + D_{mm}$, where $A_{m \cdot}$ denotes the restriction of matrix $A = A_{n \times l}$ to the rows indexed by $m$.

DMFA is trained to minimize the negative log-likelihood of conditional density $p(x_m|x_o)$ which is given by:

$$l(x_o, x_m) = -\log p(x_m|x_o) = -\log N(\mu_m, \Sigma_{mm})(x_m).$$

Observe that the above objective is calculated only on the parameters of $\mu, \Sigma$ corresponding to missing values (other entries are not used by the model). This means that $f$ can theoretically return irrelevant values on coordinates related to the observed values. The most important thing is that DMFA directly minimizes the log-likelihood of $p(x_m|x_o)$ and thus should provide a better estimate of missing values than using conditional density obtained by a typical GMM.

Let us highlight that we do not need to specify the number of mixture components as in the classical GMM. Once the neural network is fed with a missing data point, it generates an individual density for this data point. In the case of the classical mixture model, conditional density is formed by restricting the most probable Gaussian components (from the set of mixture components) to missing values. In consequence, our conditional density should be more expressive than the one obtained from the classical GMM, where the number of components is fixed.

## 3. Experiments

In this section, we compare the quality of a density produced by DMFA with a conditional density obtained from GMM. We intentionally do not use methods based on deep generative models (mentioned in the introduction), because they do not give a closed-form expression for a conditional density. We consider two typical image datasets: MNIST (LeCun et al., 1998) and CelebA (Liu et al., 2015) (aligned, cropped and resized to $32 \times 32$) datasets. The code is available at Github: `https://github.com/mprzewie/dmfa_inpainting`.

DMFA is instantiated using 4 convolutional layers. This is followed by a dense layer, which produces the final output vectors (the number of latent dimensions $l$ determining the covariance matrix equals 4). Our model is trained with a learning rate $4 \cdot 10^{-5}$ for 50 epochs. As a baseline, we use the implementation of MFA (Richardson & Weiss, 2018) trained in a classical way[2]. Following the authors' code, the number of components $k$ and latent dimensions $l$ equal: $k = 50, l = 6$ for MNIST and $k = 300, l = 10$ for CelebA.

We examine the imputation constructed by replacing missing values with the mean vector of corresponding conditional density. For each test image, we drop a square patch,

---

[2]The code was taken from `https://github.com/eitanrich/torch-mfa`.

image input MFA DMFA image input MFA DMFA

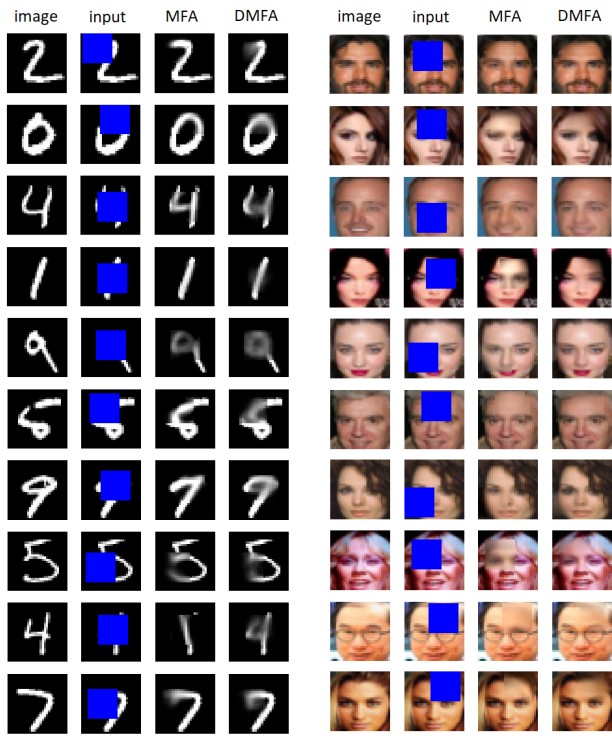

*Figure 2.* Sample imputation results produced by DMFA and MFA.

*Table 1.* Negative log-likelihood (NLL) and mean-square error (MSE) of the most probable imputation obtained by DMFA and MFA (lower is better).

| Dataset | Measure | MFA | DMFA |
|---------|---------|------|---------|
| MNIST | NLL | 58.10 | -244.81 |
| | MSE | 18.59 | 12.96 |
| CelebA | NLL | -882.54 | -1222.85 |
| | MSE | 9.82 | 7.73 |

*Table 2.* Negative log-likelihood (NLL) and mean-square error (MSE) of the most probable imputation obtained by DMFA and MFA when the sizes of missing regions were different in training and test set (test set contains the missing regions of the size: 10x15 for MNIST and 12x20 for CelebA).

| Dataset | Measure | MFA | DMFA |
|---------|---------|---------|---------|
| MNIST | NLL | 5.85 | -147.29 |
| | MSE | 12.09 | 9.04 |
| CelebA | NLL | -815.60 | -885.85 |
| | MSE | 11.73 | 10.03 |

which covers 1/4 area of image. Analogical missing regions were generated for training images. Sample results presented in Figure 2 for MNIST show that MFA produces sharper imputations than DMFA. However, the results returned by MFA do not always agree with ground-truth (7th and 9th rows). This is confirmed by verifying the mean-square error (MSE) of imputations, Table 1. Since DMFA usually gives images more similar to the ground-truth, it obtains lower MSE values than MFA. It is also evident from Table 1 that a density returned by DMFA has significantly higher log-likelihood, which means that DMFA finds a better solution to the underlying problem. In the case of CelebA, DMFA produces more visually plausible imputations, which at the same time coincides with ground-truth. The results obtained by MFA are not satisfactory. The visual inspection is confirmed by quantitative assessment presented in Table 1.

In previous experiments, DMFA was trained on the same sizes of missing regions as they appear in the test set. We check whether DMFA can deal with estimating conditional distributions on missing patterns that was not presented in the training set. For this purpose, we modified a test set and created missing patterns of the size: 10x15 for MNIST and 12x20 for CelebA. The results presented in Table 2 show that DMFA still gives better performance that MFA both in terms of NLL and MSE. However the difference between these models is smaller than before in the case of CelebA.

It suggests that creating more diverse missing patterns in training set could improve the performance on test set in this case. Note however that MFA does not use missing values in training and, in consequence, it was supposed to perform better in this case. Visual comparison is shown in the Figure 3.

## 4. Conclusion

We proposed a deep learning approach for estimating the conditional Gaussian density of missing values given the observed ones. Experiments showed that the obtained density has significantly lower value of negative log-likelihood function than conditional GMM trained in a classical way. Moreover, imputations produced by replacing missing values with the mean vector of resulting Gaussian look visually plausible.

## Acknowledgements

This work was partially supported by the National Science Centre (Poland) grant no. 2018/31/B/ST6/00993 and 2017/25/B/ST6/01271.

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

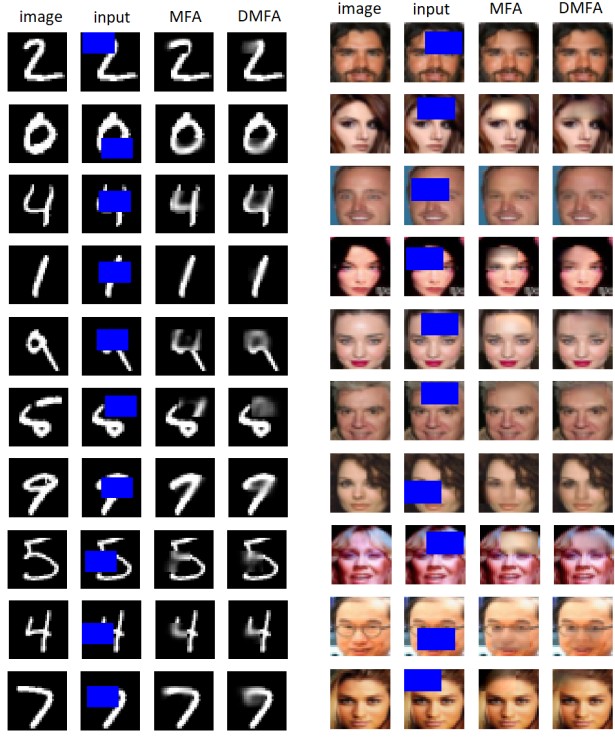

Figure 3. Sample imputation results produced by DMFA and MFA when the sizes of missing regions were different in training and test set.

*Single-channel recording*, pp. 589–633. Springer, 1995.

Delalleau, O., Courville, A., and Bengio, Y. Efficient em training of gaussian mixtures with missing data. *arXiv preprint arXiv:1209.0521*, 2012.

Dick, U., Haider, P., and Scheffer, T. Learning from incomplete data with infinite imputations. In *Proceedings of the 25th international conference on Machine learning*, pp. 232–239, 2008.

Ghahramani, Z. and Jordan, M. I. Supervised learning from incomplete data via an em approach. In *Advances in neural information processing systems*, pp. 120–127, 1994.

Ghahramani, Z., Hinton, G. E., et al. The em algorithm for mixtures of factor analyzers. Technical report, Technical Report CRG-TR-96-1, University of Toronto, 1996.

Goodfellow, I., Pouget-Abadie, J., Mirza, M., Xu, B., Warde-Farley, D., Ozair, S., Courville, A., , and Bengio, Y. Generative adversarial nets. In *Advances in Neural Information Processing Systems*, pp. 2672–2680, 2014.

Goodfellow, I., Bengio, Y., and Courville, A. *Deep learning*. MIT press, 2016.

Iizuka, S., Simo-Serra, E., and Ishikawa, H. Globally and locally consistent image completion. *ACM Transactions on Graphics (ToG)*, 36(4):1–14, 2017.

Jerez, J. M., Molina, I., García-Laencina, P. J., Alba, E., Ribelles, N., Martín, M., and Franco, L. Missing data imputation using statistical and machine learning methods in a real breast cancer problem. *Artificial intelligence in medicine*, 50(2):105–115, 2010.

Kingma, D. and Welling, M. Auto-encoding variational Bayes. In *International Conference on Learning Representations*, 2014.

LeCun, Y., Bottou, L., Bengio, Y., and Haffner, P. Gradient-based learning applied to document recognition. *Proceedings of the IEEE*, 86:2278–2324, 1998.

Li, S. C.-X., Jiang, B., and Marlin, B. Misgan: Learning from incomplete data with generative adversarial networks. *arXiv preprint arXiv:1902.09599*, 2019a.

Li, Y., Akbar, S., and Oliva, J. B. Flow models for arbitrary conditional likelihoods. *arXiv preprint arXiv:1909.06319*, 2019b.

Liu, Z., Luo, P., Wang, X., and Tang, X. Deep learning face attributes in the wild. In *International Conference on Computer Vision*, 2015.

Ma, C., Gong, W., Hernández-Lobato, J. M., Koenigstein, N., Nowozin, S., and Zhang, C. Partial vae for hybrid recommender system. In *NIPS Workshop on Bayesian Deep Learning*, 2018.

Mattei, P.-A. and Frellsen, J. Leveraging the exact likelihood of deep latent variable models. In *Advances in Neural Information Processing Systems*, pp. 3855–3866, 2018.

Mattei, P.-A. and Frellsen, J. Miwae: Deep generative modelling and imputation of incomplete data sets. In *International Conference on Machine Learning*, pp. 4413–4423, 2019.

McLachlan, G. J. and Peel, D. *Finite mixture models*. John Wiley & Sons, 2004.

Mirza, M. and Osindero, S. Conditional generative adversarial nets. *arXiv preprint arXiv:1411.1784*, 2014.

Nazabal, A., Olmos, P. M., Ghahramani, Z., and Valera, I. Handling incomplete heterogeneous data using vaes. *Pattern Recognition*, pp. 107501, 2020.

Ng, A. Y. and Jordan, M. I. On discriminative vs. generative classifiers: A comparison of logistic regression and naive bayes. In *Advances in neural information processing systems*, pp. 841–848, 2002.

Papamakarios, G. and Murray, I. Fast $\varepsilon$-free inference of simulation models with bayesian conditional density estimation. In *Advances in Neural Information Processing Systems*, pp. 1028–1036, 2016.

Pathak, D., Krahenbuhl, P., Donahue, J., Darrell, T., and Efros, A. Context encoders: Feature learning by inpainting. In *IEEE Conference on Computer Vision and Pattern Recognition*, pp. 2536–2544, 2016.

Rezende, D. J., Mohamed, S., and Wierstra, D. Stochastic backpropagation and approximate inference in deep generative models. *arXiv preprint arXiv:1401.4082*, 2014.

Richardson, E. and Weiss, Y. On gans and gmms. In *Advances in Neural Information Processing Systems*, pp. 5847–5858, 2018.

Śmieja, M., Struski, Ł., Tabor, J., Zieliński, B., and Spurek, P. Processing of missing data by neural networks. In *Advances in Neural Information Processing Systems*, pp. 2719–2729, 2018.

Śmieja, M., Struski, Ł., Tabor, J., and Marzec, M. Generalized rbf kernel for incomplete data. *Knowledge-Based Systems*, 173:150–162, 2019.

Sohn, K., Lee, H., and Yan, X. Learning structured output representation using deep conditional generative models. In *Advances in neural information processing systems*, pp. 3483–3491, 2015.

Tipping, M. E. and Bishop, C. M. Probabilistic principal component analysis. *Journal of the Royal Statistical Society: Series B (Statistical Methodology)*, 61(3):611–622, 1999.

Tolstikhin, I., Bousquet, O., Gelly, S., and Schölkopf, B. Wasserstein auto-encoders. arXiv:1711.01558, 2017.

Trippe, B. L. and Turner, R. E. Conditional density estimation with bayesian normalising flows. *arXiv preprint arXiv:1802.04908*, 2018.

Van Buuren, S. *Flexible imputation of missing data*. CRC press, 2018.

Van den Oord, A., Kalchbrenner, N., Espeholt, L., Vinyals, O., Graves, A., et al. Conditional image generation with pixelcnn decoders. In *Advances in neural information processing systems*, pp. 4790–4798, 2016.

Williams, D. and Carin, L. Analytical kernel matrix completion with incomplete multi-view data. In *Proceedings of the International Conference on Machine Learning (ICML) Workshop on Learning with Multiple Views*, pp. 80–86, 2005.

Yoon, J., Jordon, J., and Van Der Schaar, M. Gain: Missing data imputation using generative adversarial nets. *arXiv preprint arXiv:1806.02920*, 2018.

Yu, J., Lin, Z., Yang, J., Shen, X., Lu, X., and Huang, T. S. Generative image inpainting with contextual attention. In *IEEE Conference on Computer Vision and Pattern Recognition*, pp. 5505–5514, 2018.
