# OpenReview forum: "Estimating conditional density of missing values using deep Gaussian mixture model"
_ICML.cc/2020/Workshop/Artemiss — ICML Artemiss 2020_

### Official Review · AnonReviewer1 · 2020-07-01
**Clearly written paper with some issues**

**Confidence:** 5
**Rating:** 6

**Review:**

The paper proposes a discriminative method for learning the conditional density of missing values given observed values for incomplete data using a deep Gaussian mixture model. Conceptually, the proposed method works as follows: the observed part of an incomplete data point is fed into a neural network that outputs the parameters of a Gaussian distribution (parametrised as a factor analysis model). This Gaussian distribution represents the conditional density of the missing values given the observed values. *Essentially, the objective of the model is to learn all the conditionals of $p(x)$*. The model is learned by maximising the conditional likelihood function. Experiments are performed on MNIST and CelebA, where for each test image a square patch covering 1/4 of the image is missing. The results show that the proposed model compares favourably to a Mixture of Factor Analyzers model.

The paper is straight forward to follow and appears to be scientifically sound. However, there is a number of issues with the paper, which places the paper on the borderline between accept and reject:

(1) The authors have decided to approach the problem in a discriminative fashion; on page three, they argue:

> "The most important thing is that DMFA directly minimizes the log-likelihood of $p(x_m |x_o )$ and thus should provide a better estimate of missing values than using conditional density obtained by a typical GMM."

First of all, this statement should be back up by a citation (e.g. Ng and Jordan, 2002). Second of all, the number of possible conditionals of $p(x)$ grows exponentially with the dimension of $x$, e.g. for MNIST there are $2^{32\times32}$ conditionals. It appears to be computationally infeasible to learn all these conditionals and thereby have a good model for an arbitrary missing pattern. The authors should discuss why they think it is feasible learning all the conditionals of $p(x)$ and experimentally verify how well the model performs when it a test time is presented with a conditional that was not in the training set.

(2) VAE has, from their inception, been used for imputing missing values. In the original paper by Rezende et al. (2014), a joint VAE $p(x)$ is learned on complete data, and for incomplete data at test time, this model is used for imputing missing values by sampling from $p(x_m|x_o)$ using pseudo-gibbs sampling (more advanced imputation schemes were discussed by Mattei and Frellsen, 2018). Furthermore, there is a vast literature on how deep generative models can be learned, when the training data is also incomplete by maximising $\log p(x_o)$, see e.g. Nazabal et al. (2018), Ma et al. (2018), and Mattei and Frellsen (2019). The authors should discuss this related literature, and also compare to these methods; e.g. they could compare the imputation of DMFA to the imputations obtained using a VAE.

(3) It appears that the DMFA model only can be trained on complete data. If this is the case, it should be clearly stated in the paper. Furthermore, it should be explained which conditional are learned in the training process (i.e. which missing patterns are artificially introduced in the training data). Also, the authors should also mention which type(s) of missing process(s) they are considering, i.e. MCAR, MAR or MNAR.

### Minor comments:

* In figure 1: "Given a missing data point, our model returns the parameters of conditional Gaussian density [...]" should be "Given the observed values [...]".

* On page two the authors write that the dataset is $X \subset \mathbb{R}^n$. This obviously a misnomer.

### Reference

C. Ma, W. Gong, J. M. Hernández-Lobato, N. Koenigstein, S. Nowozin, and C. Zhang. Partial VAE for hybrid recommender system. NIPS Workshop on Bayesian Deep Learning, 2018.

P.-A. Mattei and J. Frellsen. Leveraging the exact likelihood of deep latent variable models. In Advances in Neural Information Processing Systems, pp. 3855–3866, 2018

P.-A. Mattei and J. Frellsen. MIWAE: Deep generative modelling and imputation of incomplete data sets. In International Conference on Machine Learning, pp. 4413–4423, 2019.

A. Nazabal, P. M. Olmos, Z. Ghahramani, and I. Valera. Handling incomplete heterogeneous data using VAEs. arXiv:1807.03653, 2018.

A. Y. Ng, and M. I. Jordan. On discriminative vs. generative classifiers: A comparison of logistic regression and naive bayes. In Advances in neural information processing systems, pp. 841-848, 2002.

D. Rezende, S. Mohamed, and D. Wierstra. Stochastic backpropagation and approximate inference in deep generative models. In Proceedings of the 31st International Conference on Machine Learning, pp. 1278–1286, 2014.

---

### Decision · Program_Chairs · 2020-07-02

**Decision:**

Accept

**Comment:**

We're happy to accept this paper at Artemiss. We'll contact you soon to inform you about more details concerning the format of your presentation at the workshop, and the camera-ready version deadline. Please take into account the referee's comments to write the camera-ready version.